# Extracellular Vesicle Identification Using Label-Free Surface-Enhanced Raman Spectroscopy: Detection and Signal Analysis Strategies

**DOI:** 10.3390/molecules25215209

**Published:** 2020-11-09

**Authors:** Hyunku Shin, Dongkwon Seo, Yeonho Choi

**Affiliations:** 1Department of Bio-convergence Engineering, Korea University, Seoul 02841, Korea; ramgee91@gmail.com (H.S.); ss4827@korea.ac.kr (D.S.); 2School of Biomedical Engineering, Korea University, Seoul 02841, Korea

**Keywords:** extracellular vesicles, surface-enhanced Raman spectroscopy, nanostructures, signal analysis

## Abstract

Extracellular vesicles (EVs) have been widely investigated as promising biomarkers for the liquid biopsy of diseases, owing to their countless roles in biological systems. Furthermore, with the notable progress of exosome research, the use of label-free surface-enhanced Raman spectroscopy (SERS) to identify and distinguish disease-related EVs has emerged. Even in the absence of specific markers for disease-related EVs, label-free SERS enables the identification of unique patterns of disease-related EVs through their molecular fingerprints. In this review, we describe label-free SERS approaches for disease-related EV pattern identification in terms of substrate design and signal analysis strategies. We first describe the general characteristics of EVs and their SERS signals. We then present recent works on applied plasmonic nanostructures to sensitively detect EVs and notable methods to interpret complex spectral data. This review also discusses current challenges and future prospects of label-free SERS-based disease-related EV pattern identification.

## 1. Introduction

Extracellular vesicles (EVs) are secreted by most cells and contain cytosolic and membrane substances of their parent cells [1,2,3]. They include exosomes, ectosomes, and exomeres. The cargo and membrane substances of EVs can provide useful clues regarding the in vivo status [4,5,6]. Thus, EVs are widely investigated and have been highlighted as a promising biomarker for various diseases, including cancers [7]. To analyze EVs, various techniques are employed, including transmission electron microscopy (TEM) [8], nanoparticle tracking analysis [9,10], and surface plasmon resonance sensors [11,12,13].

Raman spectroscopy is an optical and analytical technique that can offer useful information about EVs. Raman spectroscopy has been used to observe the inelastic scattering of light from analytes using laser irradiation [14]. Molecules within the focused volume of the laser produce narrow and sharp bands that are related to the chemical bonds of their organic compounds. Raman spectroscopy has excellent advantages in non-destructive and label-free methods, but it also suffers from the critical drawback of delivering extremely low signal intensity [15]. SERS is a powerful method to overcome this obstacle and enables the identification of molecules by intensively enhanced electromagnetic (EM) fields within metallic nanogaps [16]. When laser light irradiates the metallic nanogap substrate, the Raman spectrum of molecules at the proximity of the nanogap can be highly amplified up to an order of 10^8^ [17]. Since SERS signals offer molecular fingerprints of unique chemical structures, many research groups have reported SERS-based detection of EVs [18,19,20,21,22,23,24,25,26,27,28,29,30,31].

Typically, SERS-based approaches can be categorized into indirect and direct [32,33]. The indirect SERS relies on chromophore-based Raman dyes that produce unique and strong intensities at certain spectral bands [34]. Typically, the Raman dyes are tagged with the hotspot of plasmonic nanoparticles, and these nanoparticles are induced to attach to the target molecules using specific linkers such as antibodies [35,36,37] and aptamers [38,39,40]. This approach selectively identifies the target molecules by monitoring the emitted light from the Raman dyes. Owing to its high sensitivity, indirect SERS has been widely implemented in the detection of biomolecules such as EVs [18,19,20,21,22,23,24,25,26,27]. Even though genetic and molecular profiling of disease-related EVs has been continuously investigated, unfortunately specific and effective markers of disease-related EVs have not been found to be evident in many cases [41,42,43,44]. Thus, tracking disease-related EVs using a specific linker is quite challenging.

In contrast, direct SERS detects the intrinsic Raman spectrum of the analyte and is also called label-free SERS because it does not require any labeling using Raman dyes [32,45,46,47]. This approach makes it possible to directly detect molecular fingerprint signals that originate from the surface molecules of EVs [48]. Many research groups utilizing label-free SERS to analyze EVs focus on detecting intrinsic signals of the target, ultimately aiming to identify disease-related EVs [29,49,50,51,52,53,54,55,56,57,58,59]. Label-free SERS has several advantages. First, it does not require a specific marker. Given that label-free SERS explores signal patterns that can be derived from either undiscovered or from substances of no interest, it can analyze EVs that are difficult to distinguish by other analytical methods targeting a unique marker [60,61]. Second, the Raman spectrum contains a large amount of information regarding detected EVs. Because the scattering bands in the EV-derived Raman spectrum present information regarding the chemical structure, it can provide clues about the molecular composition of EVs [55].

Typically, label-free SERS approaches for biomolecules can be divided into two major parts: (1) detection of SERS signals and (2) analysis of the signals. In this review, we summarize recent studies concerning label-free SERS for EV analysis in terms of SERS substrate and signal analysis strategies (Figure 1). First, we describe the background of EVs and the characteristics of their SERS signals. Then, we summarize the recent methodologies for nanostructures and signal analysis for the label-free SERS of EVs. Moreover, challenges and prospects associated with label-free SERS are also discussed.

## 2. Characteristics of EVs and Their SERS Signals

Since EVs were discovered in 1983, they have been one of the most widely investigated extracellular vesicles [62]. EVs have a size of 30–150 nm, arise from the multivesicular bodies in a cell, and have a phospholipid bilayer [2]. EVs play countless roles in cell-to-cell interactions and in the transfer of biochemical substances from parent cells to other cells [63]. Many studies have reported that EVs are associated with the mechanisms of cancer metastasis [64], tumor growth [65], and immune activation [63]. EVs exist in biological fluids, such as blood, urine, and saliva. Thus, they have been of great interest as potential biomarkers for liquid biopsies [66,67,68].

EVs have been widely investigated, given that they contain many biomolecules, including DNA, miRNA, proteins, and lipids. In particular, the surface composition of EVs is of special interest. Unique substances on the surface of EVs facilitate their non-destructive capture and close examination. Furthermore, the EV surface is coated with dense glycans attached to surface proteins and outer lipids [69,70]. Recently, it has been reported that monitoring of surface glycans can be used to identify EVs from different biological events [70]. Membrane proteins are one of the most significantly investigated substances on the surface of EVs. Escola et al. [71] reported that several tetraspanin proteins are enriched in EVs. Additionally, CD81, CD9, and CD63 are the representative tetraspanin proteins that are highly enriched on the EV surface and are quite common as EV markers [72]. Notably, various membrane proteins that are enriched in disease-related EVs have been reported, which offer the possibility of EVs as potential prognostic and diagnostic biomarkers for diseases such as cancer [73,74,75,76,77,78]. Lipid molecules such as lysophospholipids, cholesterol, sphingolipids, and ceramides are also enriched in the EV membrane [69]. Skotland et al. [79] reported the potential use of lipid composition of urinary EVs as a biomarker for prostate cancer. Moreover, several studies have shown that nucleic acid (NA)-like DNA can be present on the surface of EVs [80,81,82]. Considering the physiological roles of EV-derived DNA, it is expected to be a useful biomarker [69].

The most straightforward method to obtain the SERS signal from EVs is to attach pure and non-destructive EVs onto SERS substrates. The signal intensity in SERS decays exponentially as an analyte drifts away from the hotspot [83]. Thus, to obtain the SERS signal sensitively, EVs must be located in proximity to a SERS hotspot. Considering the thickness of the lipid bilayer covering the EVs, a significant portion of the SERS signal may originate from the outer surface of EVs. The EV-derived SERS spectrum has bands that can be assigned to various amino acids, lipids, and NAs [29,53]. The emerging results demonstrate that label-free SERS detection of EVs makes it possible to identify a comprehensive signal from the EV-derived surface composition [55].

However, to employ label-free SERS for EV analysis, several obstacles need to be resolved. First, EVs typically produce extremely low signal intensities (Figure 2a). Because EVs are mainly composed of non-chromophore biomolecules, detecting their unique signal patterns is more challenging than in molecules with chromophores [84]. Additionally, EVs are relatively larger than SERS hotspots; thus, it is difficult to locate EVs abundantly in the SERS-effective hotspot. Accordingly, a specialized strategy for EV detection should be developed to sensitively obtain their signals. Second, the Raman spectrum of biomolecules like EVs is heterogeneous and complex; thus, it is often difficult to interpret (Figure 2b) [48,59]. The molecular fingerprint signal can involve an abundance of information regarding the biomolecule but is also a fiendish puzzle. Because EVs are larger than monomers or small biomolecules, the relative position of EVs on the surface of a metal nanostructure has an absolute effect on the signal intensity. Moreover, the complicated composition of EVs in biological samples (e.g., blood and urine) renders analysis against human samples harder. Therefore, effective methods to interpret and analyze complex and heterogeneous signals are required.

## 3. Label-Free SERS to Identify EVs

### 3.1. EV Detection Methods Using Label-Free SERS Substrate

As noted earlier, the low signal intensities and heterogeneous nature of EVs hinder label-free SERS EV detection. To overcome these problems, researchers need to place EVs onto the SERS hotspot. Accordingly, researchers have focused on inducing EVs to affix themselves to the surface of plasmonic nanoparticles or nanostructures, by utilizing the chemical and physical properties of EVs. These approaches can be categorized into two groups depending on the EV-capturing method.

#### 3.1.1. Drying-State Detection Methods

The most straightforward method for label-free SERS detection is drying EVs onto the SERS substrate. After evaporation of the solvent, the vesicles in the solution can be coated onto the substrate and captured in the proximity of the hotspot. For this purpose, Park et al. [59] reported that the coffee-ring effect induced a gold nanoparticle (AuNP)-based SERS substrate. They employed CuSO_4_ to aggregate the AuNPs to form a SERS hotspot on the substrate and then dried it. Next, an EV solution was dried on the substrate coated with aggregated AuNPs. Consequently, they sensitively obtained the intrinsic SERS signal of exosome-like EVs (ELVs). This method has been used in many studies, since it is simple and easy to use. However, owing to the non-uniformity of the hotspot due to the coffee-ring effect and position of the EVs, this method is unsuitable for uniform SERS substrates.

To solve the non-uniformity issue related to the nanoparticle-based SERS substrate, Lee et al. [29] established a drying method using a nanobowl platform substrate. The nanobowls were fabricated by employing polydimethylsiloxane using a polystyrene latex particle layer as a mold. Then, the surface of the nanobowl platform was sputtered using a silver nanofilm for plasmonic effects. This metallic nanobowl structure has two advantages: (1) The nanobowl platform uniformly enhances the EM field inside the structure (Figure 3a); (2) by means of the petal effect, which enables the EV solution to spread over the nanobowl structures (Figure 3b). With these advantages, the nanobowl structure induces the entire vesicles to be positioned within the SERS-active area. In another study, Sivashanmugan et al. [49] fabricated a plasmonic gap-mode SERS substrate using an Au nanorod (AuNR) array of high-density hot rings. Then they captured silver nano-cubes (AgNCs) using a dithiol coating (Figure 3c). Owing to the secured AgNCs on the AuNR pattern, the substrate forms a narrow bimetallic nanogap (<2 nm). Furthermore, it increases the SERS enhancement factor by approximately 9.11 × 10^8^ times higher than that of the spontaneous Raman signal (Figure 3d). After the ELV solution was dried on the substrate, ELVs were stuck to the top of the AuNR array and bimetallic nanogap, allowing them to obtain SERS signals even from 10^4^–10^5^ times diluted solutions (the concentration of the original solution was from 1.1 × 10^6^ to 1.2 × 10^8^ particle·ml^−1^).

These studies showed that label-free SERS detection by the drying method could guarantee the amplification of the EM field for signal detection. However, randomly posited EVs and the narrow size of the EM field enhancement are obstacles to dried-state detection. In particular, the narrow amplified EM field makes it difficult to obtain the entire range of SERS signal of the EVs. Moreover, EVs must be positioned in the amplified area. To overcome these problems, the amplified EM field must be expanded or the EVs must be physically captured in the hotspot.

To generate a broadened and amplified EM field, Yan et al. [51] utilized a hybrid substrate of a graphene-covered-nanopyramid array for EV detection (Figure 4a). To verify uniformly amplified EM field generation, they simulated a graphene-covered-nanopyramid array using the finite-difference time-domain (FDTD) method. According to the simulation results, this substrate increased the EM field by more than 10^12^ times and generated a uniform EM field along the surface due to the internal EM field of graphene (Figure 4b). In addition, graphene provided another advantage. Graphene is a bioinert material; thus, it protects metallic nanostructures from corrosion, oxidation, and direct absorption of biological samples. Consequently, they successfully established SERS mapping against dried EVs suspended on the substrate, even with diluted samples (Figure 4c,d).

Moreover, a novel strategy for trapping EVs using the physical properties of vesicles was introduced. Dong et al. [50] fabricated an Au-coated TiO_2_ beehive structure for trapping and measuring EVs. They trapped EVs within the structure and enhanced SERS by means of the “slow light effect,” which is a property of the TiO_2_ beehive structure. The beehive structure (known as macroporous inverse opal; MIO) absorbs sound waves by means of multi-scattering processes. Moreover, TiO_2_ is an anisotropic material with a high refractive index and has high scattering properties that allow the trapping of light. Additionally, TiO_2_ beehive structures have an optical stopband that traps light inside the pores. Therefore, the slow light effect enhances the SERS signal of not only small molecules but also large particles, which is quite different from the general EM field enhancement. Thus, Dong et al. attempted to optimize the TiO_2_ MIO structure for suitable SERS applications. For this, they fabricated TiO_2_ MIO structures of various sizes (Figure 5a) and confirmed that the 290 nm TiO_2_ MIO structure effectively traps light and enhances the EM field through near-field and FDTD simulations (Figure 5b,c). Furthermore, they compared the methylene blue-specific SERS peak (at 1623 cm^−1^) intensity to the optimized thickness of the gold film (Figure 5d). Consequently, the SERS effect increased by more than 1.25 × 10^3^ times compared to that of the spontaneous Raman effect when using 80 nm Au-coated 290 nm TiO_2_ MIO. Moreover because of its beehive-like structure, MIO can isolate EVs from biofluids using its nanopores (Figure 5e), making it possible to obtain label-free EV-derived SERS signals without further EV isolation.

This type of method using physical trapping has been rarely reported, but it has potential for the effective detection of EVs. Label-free SERS approaches for vesicle-like analytes, including viruses and spherical nanoparticles, have been continuously reported. These methods can be adopted for EV detection to significantly improve signal sensitivity [85,86].

The SERS techniques mentioned above obtained the SERS signal by drying the EV solution on the substrate. This method can place EVs near the SERS hotspot without any chemical processes. However, this study has several limitations. The drying process involving solvent evaporation is usually time-consuming. Additionally, EVs can be damaged during the drying process because of osmotic shock. Furthermore because the entire substance in the solution can be concentrated and coated onto the substrate, SERS signals of impurities can also be detected and produce undesired signals.

#### 3.1.2. Liquid-State Detection Methods

For fast and non-destructive detection of EVs, some research groups have attempted to obtain EV-derived SERS signals in the liquid state. To obtain such signals in the liquid state, it is necessary to capture EVs near the SERS hotspot with ligands or linkers. Since phospholipids and various surface proteins cover EVs, they usually have a negatively charged surface [87,88]. Therefore, several researchers have used chemical ligands or linkers, such as antibodies that bind to phospholipids or surface proteins to form the EVs-SERS substrate complex.

Among the studies using ligands, Stremersch et al. [56] used positively charged AuNPs coated with 4-dimethyl aminopyridine (DMAP). The DMAP acts as a cationic substance on the AuNP surface, thereby absorbing the negatively charged EV surface. Because of the absorption of DMAP-coated AuNPs, the surface of EVs covered by DMAP-coated AuNPs resulted in the formation of DMAP, in order to mediate AuNPs-EV complexes. Moreover, to ensure signal uniformity and effective SERS hotspot formation, the absorption was controlled by adjusting the AuNP/EV ratio (Figure 6a). Notably, they were able to effectively coat AuNPs on individual EVs without aggregation by controlling the ratio of AuNP to EV. TEM results showed that DMAP-AuNPs covered over 40% of the EV surface at an optimal ratio. They then obtained an individual EV-derived SERS signal. Although the strongest signals were predominantly derived from capping DMAP on the AuNP surface, several peaks originating from EVs were found in the EV samples, indicating the viability of the single vesicle SERS detection (Figure 6b). However, the capping reagents can interfere with the signal because they mediate the connection between the SERS hotspot and the vesicles (Figure 6c). Thus, the same research group also demonstrated a method to overcome the problem of capping reagents [57]. To achieve this, they replaced DMAP on the EV surface with silver through reduction and consequently succeeded in inducing closer access to the plasmonic surface and the lipid bilayer. According to the simulation and electric field enhancement factor (EFEF) calculation, they expected a 6.9-times improved SERS signal sensitivity (Figure 6d). Additionally, they confirmed that distinct EV-derived SERS signals were obtained compared to the typical ligand-based nanoparticles (Figure 5e). These methods take only 0.5 s to obtain the SERS signal from an individual EV, owing to the significantly enhanced EM field around the EV surface.

In addition to inducing nanoparticles to be adsorbed on vesicles, a method was also introduced to induce vesicles to be coated on a substrate [55]. To achieve this, AuNP-aggregated substrates coated with cysteamine were employed. Furthermore, given that cysteamine has a thiol and an amine group in its structure, it can bind to the Au surface using the thiol group and induce electrostatic binding with negatively charged vesicles using the amine group. After exosome treatment of the substrate, EV-derived SERS signals could be obtained in the liquid state. These assays were faster than the dry state detection method and did not destroy the EVs, so that a stable signal of the surface protein could be obtained. In particular, Stremersch’s et al. method showed the advantage of obtaining the entire EV surface signal because of the surface being covered with nanoparticles. However, it has not been able to solve the heterogeneity issue caused by the signal difference between individual EVs. This problem arises because of the existence of multiple EVs in the SERS measurement area. To solve this problem, it is necessary to acquire a clear SERS signal through stronger amplification as well as trapping of individual EVs using external force.

Despite the abovementioned efforts, it is difficult to clearly identify the difference in signals of EVs because of the interference from various substances, like proteins, nucleotides, and phospholipids, during a simple peak analysis of the SERS signal. Therefore, to use the SERS signal of EVs as a fingerprint for disease diagnosis, a more precise signal analysis method is required.

### 3.2. Signal Analysis of the Label-Free SERS of EVs

#### 3.2.1. Conventional Method

The most conventional approach to interpret the SERS signal of EVs is to explore specific Raman bands and analyze recognizable differences in terms of intensity and Raman shift when compared to a control group. Typically, since biological analytes, including EVs, differ in the organic compositions that consist of proteins, lipids, and NAs, the difference in the fingerprint region from 400 to 1800 cm^−1^ is observed.

Sivashanmugan et al. [49] focused on meaningful differences in Raman bands between healthy cells (NL-20, BEAS-20, and L929)-derived normal EVs and lung cancer cells (PC-9, H1975, and HCC827)-derived EVs. All normal lung EVs exhibited strong peaks in the Raman spectra at 625, 1254, and 1404 cm^−1^ (Figure 7a). These peaks can be assigned to C-C twisting, amide III, phosphate stretching, and CH rocking, related to proteins, lipids, and NAs. Additionally, they reported a comprehensive assignment associated with the numerous major bands observed from EVs, which are derived from each cancer cell line. These results confirm that the intrinsic SERS signals of cancerous EVs may be different because of different biomolecule compositions.

Since the specific Raman bands are deeply correlated with molecular bonds in an analyte, it is imperative to identify the status of a certain protein. Recently, Dong et al. [50] detected bands associated with P-O bonds by phosphoproteins in cancerous EVs through label-free SERS. The protein phosphorylation status of RWPE-1 cell (Healthy cell)- and LNCaP cell (prostate cancer cell)-derived EVs were analyzed. They first established the specific signal of phosphorylated protein and determined the strong intensity at 1087 cm^−1^ assigned to the P-O bond. Notably, in a comparative study of EV signals, LNCaP cell-derived exosomes showed high signal intensity at the peaks (Figure 7b). Interestingly, in EVs derived from other cancer cell lines, the signal intensity related to protein phosphorylation was stronger than their corresponding normal cell line EVs. This indicates that protein phosphorylation in cancerous EVs is more abundant than in healthy cell-derived normal EVs, and comparing intensities at the Raman band can allow us to identify cancerous EV samples. Moreover, comparative analysis between ten healthy controls and 15 prostate cancer patients showed that the EVs of all healthy controls had a low signal intensity at 1087 cm^−1^ (Figure 7c). Notably, EVs of other cancer patients, including lung, liver, and colon cancer, also have a relatively high protein phosphorylation status. Although a correlation between this signal and cancer-specific antigens has not been found, it is significant that qualitative analysis of Western blotting can be improved by quantitative analysis with label-free SERS.

Conventional methods employing the assignment of specific bands of EVs have been able to provide useful information regarding specific molecular structures that lead to differences between EVs. However, in SERS measurements, the fluctuation of the Raman signal intensity can cause a problem of poor reproducibility [89]. In particular because the molecular orientations and positions in the SERS hotspot are closely related to the generated signal intensity, a heterogeneous signal pattern may be produced [89]. Additionally, even a slight difference in SERS substrates can generate large fluctuations in both peak position and signal intensity [90,91]. Therefore, statistical methods have been applied to derive differences across an entire spectral range rather than using a specific signal band.

#### 3.2.2. Principal Component Analysis (PCA)

Because the spectrum can be regarded as a multidimensional datum having intensity values along the range of the Raman shift, multivariate statistical methods can be utilized for interpretation. PCA is a classical multivariate statistical method to analyze spectral data such as Fourier-transform infrared spectroscopy and Raman spectroscopy [92,93,94]. Typically, PCA has been employed for dimensionality reduction and feature extraction [95]. With PCA, principal components (PCs) can be extracted from the spectral data in the order of highest covariance. If the difference between other data groups is significant, the variation of data distribution along the PC axis can be maximized for each group. In the case of high-dimensional data such as a spectrum, it is possible to visualize the observed spectra in 2D or 3D through a score vector, which is dimensionally reduced through PCA. In general, data with significant differences can be clustered into independent groups. Because PCA is a linear transformation, it provides valuable information about the dominant pattern of the clusters. Park et al. [59] distinguished SERS signals of the normal alveolar cell line- and lung cancer cell line-derived EVs through PCA and investigated cancerous EV-specific signal patterns for lung cancer EV diagnosis. The EV SERS signals from 470 to 1800 cm^−1^ were clustered. In the PCA score plot, control (Bare SERS substrate), normal alveolar cell EVs, and lung cancer cell EVs were distinguished. Lung cancer EVs were classified based on PC1, with the largest variance (Figure 8a). In cross-validation, this approach showed outstanding classification performance with a sensitivity of 95.3% and specificity of 97.3%. To examine which spectral ranges affected the classification, investigation of PC loading data was performed. As a result, several spectral ranges dominant to the cancerous EVs and their corresponding normal EVs were demonstrated (Figure 8b). The dominant peaks were associated with NAs and membrane proteins. Furthermore, to test unknown samples, independent cell EV data were projected onto the PCA score plot. By evaluating the location of new data on the score plot, a sensitivity of 85.7% and a specificity of 90.0% was obtained. However, clinical samples tended to be located in the middle plane. The authors discussed the reason for the diverse origins and different proportions of EVs in the clinical samples.

In another study on cancer diagnosis using a combination of PCA and EV-derived SERS signals, Ferreira et al. [52] demonstrated the classification of nontumorigenic breast epithelium- and breast cancer-derived EVs. By using PCA, they classified SERS signals obtained from phosphate buffered saline, cancer, and non-tumoral cell EVs. To prove the concept of real-time diagnosis, tumoral/non-tumoral samples were analyzed, and both samples belonged to the reasonable 95% confidence ellipses, suggesting feasibility for real-time diagnosis (Figure 8c). PCA can also be used for the discrimination of different samples by EV signals. Yan et al. [51] reported that PCA of SERS could discriminate EVs originating from different biological sources. EV data were collected from four different sources, including lung cancer cell lines (HCC827, H1975), human serum, and fetal bovine serum. The data were clustered into different groups with a sensitivity of over 84%. These studies showed the practical results of classifying EV samples by PCA. They established sensitivity and specificity through 95% confidence ellipses bordering biologically different groups, suggesting a successful prediction for newly collected samples.

PCA can be strengthened through a combination of supervised analytical methods. For example, linear discriminant analysis (LDA) is a widely applied framework, and the PCA plus LDA approach is one of the popular combinations in many classification problems [96,97,98]. PCA is used for dimensionality reduction prior to LDA, which can reduce the computational difficulty of LDA effectively. Zhang et al. [53] demonstrated the discrimination of different cancer cell-derived EVs by PCA and LDA. Similar to other studies, different cancer cell-derived EVs were clustered in the PCA score plot. When PCA was applied to a local range of 600–760 cm^−1^ Raman shift, the esophageal cancer EVs were distinguishable and identified by LDA, showing a prediction accuracy of 97.1%. For the same approach using 940–1100 cm^−1^, breast cancer EVs were also classified with 90.6% accuracy. The classification accuracy of the EV signals in the range of 500–1600 cm^−1^ reached 96.7% and sensitivity above 95% in order to distinguish the different EV samples. In a different study, Carmicheal et al. [54] reported an EV SERS application using PCA and differential function analysis. First, the original spectra of 1004 variables (719–1800 cm^−1^) were dimensionally reduced to 20 PCs. Then, the discriminant function analysis (DFA) allows the classification of the reduced PCs into independent categories depending on the similarities and differences [99]. For classification, 121 data were collected from pancreatic epithelial cells (HPDE) and pancreatic cancer cells (MiaPaCa and CD18/HPAF). The classification efficiency in PCA was not significant, but the control group was clearly separated from the EV groups in the PC-DFA plot (Figure 8c). Additionally, individual EV groups formed discrete clusters; thus, each cell EV data were well classified with an accuracy of 90.0%. In the classification of EVs derived from normal cell lines and cancer cell lines, the sensitivity and specificity reached 90.6% and 97.1%, respectively. Additionally, EVs isolated from ten healthy individual serum controls and ten pancreatic cancer patients were also analyzed using the pre-trained PC-DFA algorithm with cell line EVs. However, the prediction accuracy was moderate compared to the cell line EV data owing to the diverse normal EVs in the patient sample. Nevertheless, this study suggested the feasibility of EVs-SERS and statistical analysis as a novel cancer detection method.

#### 3.2.3. Partial Least Square Discriminant Analysis (PLS-DA)

PLS-DA was introduced as a powerful method to analyze the SERS signals of EVs. Similar to the discrimination methods combined with PCA described earlier, PLS-DA is applied for dimensionality reduction and discriminant analysis of multivariate data [100]. PLS-DA has been widely applied to spectral data and has often reported powerful performances [100,101,102,103]. Stremersch et al. [56] demonstrated a quantification method that distinguished cancerous EV-like vesicles using PLS-DA. SERS signals of EV-like vesicles isolated from B16F10 melanoma and red blood cells (RBCs) were analyzed and discriminated by PLS-DA with high sensitivity and specificity.

Interestingly, the combination of PLS-DA and SERS showed reasonable quantification performance in mixed samples. By SERS detection of individual vesicles in two different mixtures of B16F10 and RBC vesicles in different portions, 38% and 6.3% of cancerous vesicles were predicted, respectively. This value was reasonably consistent with the proportions of 51% and 15% verified through a fluorescence method, suggesting the promising potential of label-free SERS for actual biomedical diagnosis.

## 4. Challenges and Prospects of Label-Free SERS for EVs

### 4.1. Challenges of Label-Free SERS for EVs

As discussed above, the detection method using EV capture and a statistical analysis method using PCA and PLS-DA increased the feasibility of EV diagnosis using label-free SERS. In particular, in the classification and identification of single-type cell line-derived EVs, many research groups have successfully established highly precise discrimination by the distinct signal pattern of cancerous EVs. However, in actual body fluids, a large fraction of disease-related EVs are released from diseased cells. However, most of them are secreted from healthy cells, including endothelial cells, platelets, lymphocytes, and other immune cells [92,104,105,106]. Therefore, various origins of EVs lead to the diversity of EVs, complexity due to various surface proteins, and heterogeneity in individual EVs. Thus, we need to overcome the diversity, complexity, and heterogeneity of EVs for practical clinical application of the label-free SERS technique. Additionally, the difference between in vitro and in vivo sources present significant challenges in upscaling current research progress from cell line EVs to the actual clinical field [7].

### 4.2. Prospects for Label-Free SERS for EVs

#### 4.2.1. Prospect of the Detection Methods

As a result of the various studies mentioned above, the EV detection technology using label-free SERS has made a great stride. However, for the commercialization and development of precision medicine using label-free SERS technology in the EV detection field, researchers still investigate (1) integration of EV isolation and detection, (2) more stable capturing of EVs, and (3) detection of single EVs.

For these purposes, many researchers use “microfluid.” Microfluidic technology has been in the spotlight of the precision medical field because it uses a small number of samples and enables various analyses. Particularly, microfluidic technology is used in the field of EV isolation because it can control particles of various sizes in a fluid by using a dielectric force or acoustic force [107,108,109]. In addition, microfluidic technology has been developed in the form of a bio-chip and is commercially used in various diagnostic fields, such as in SERS grafting [110,111,112]. Furthermore, research to construct a portable SERS system by miniaturizing the optical system required for the detection of SERS and utilizing it in the clinical field is underway [113]. Therefore, if microfluidic technology, which involves the use of small samples, nanoparticle collection technology, and portable SERS technology, are applied to label-free SERS, it will be applicable in commercial and precision medical fields.

As a part of these “Single EV detection” is a representative EV detection technology [48,114,115,116,117]. Single EV detection is being researched to obtain signals by preventing aggregation of EVs through surface treatment or trapping of EVs in a specific area using an external force. The simplest way to detect single EVs is to use an optic tweezer. An optic tweezer can control micro- and nano-sized particles using a focused laser. Several techniques have been recently studied to obtain information on a single exosome using optic tweezers. In particular, Dai et al. obtained signals of individual EVs using a “Raman-enabled nanoparticle trapping analysis” (R-NTA) method that combines optical forceps and label-free Raman detection (Figure 9a) [117]. However, they acquired a spontaneous Raman signal, rather than performing SERS, and confirmed morphological-chemical heterogeneity through Raman spectroscopy and confirmed heterogeneous signals between EVs derived from cancer cells and healthy cells. Additionally, they obtained Raman signals of EVs derived from two different carcinoma cell lines and cell lines in which TRPP2 was knocked down (Figure 9b) and classified them through numerical analysis (Figure 9c). As a result, it was possible to form clusters expected to contain TRPP2, a specific biomarker of the HN2 cell line, and a clear difference was observed between the averages of the signals from different clusters (Figure 9d).

Another research team captured an EV in a dielectric field using a dielectric pyramid structure. The Raman signal of the single EV was detected using a strong EM field generated by a laser at the spire of the pyramid structure. However, the “single EV detection” technologies have not yet been actively studied in SERS. Therefore, it could combine with the diversity and heterogeneity of individual EVs.

#### 4.2.2. Prospects of the Analysis Methods

Due to the progress of detection strategies, studies have been successfully conducted to obtain distinct spectral patterns of EVs derived from different sources. These achievements are expected to be developed with outstanding progress in computer science for multidimensional data and machine learning techniques. Recently, research using a convolutional neural network (CNN) to classify the Raman signals of EVs was performed [118]. The CNN-based algorithm was able to classify Raman signals obtained from EVs without background signal correction. Additionally, the algorithm showed better prediction accuracy than statistical methods, such as PCA-LDA. In addition to having the advantage of high accuracy, machine learning models can solve the heterogeneity problem of SERS signals. Machine learning algorithms, such as deep learning, have been widely applied to overcome the heterogeneity issue in the analysis of biological samples [119,120]. Additionally, many studies using artificial intelligence algorithms for Raman spectra of various biomolecules have been reported [121,122,123,124,125,126]. However, an enormous amount of data are inevitably required to train the models. For actual diagnosis, it is necessary to analyze many samples of healthy controls and patients to train and verify the artificial intelligence model. However, even if several clinical samples are obtained, isolating EVs and measuring enough SERS data is an extremely time-consuming process. To overcome this issue, the analysis of SERS data of plasma EVs in clinical samples by means of a CNN algorithm trained using cell line EV data was demonstrated recently [58]. The CNN model was trained with more than a thousand SERS signals from cell line EVs and presented a quantitative similarity between 63 plasma EVs and pre-trained lung cancer cell line EVs (Figure 10a). Lung cancer patients showed a significant difference in similarity compared to the healthy control group (Figure 10b). Notably, even the samples of stage I patients were successfully discriminated, suggesting the feasibility of early cancer diagnosis (Figure 10c). Since the development and verification of artificial intelligence technologies are continuously emerging, the introduction of artificial intelligence into EVs-SERS is expected to serve as a key to solving the current problems.

## 5. Conclusions

We have reviewed recent studies and progress in EVs label-free SERS. First, we discussed the characteristics of EV-derived SERS signals and current issues for detection and signal analysis. Then, we summarized the EV detection strategy for acquiring label-free SERS signals of EVs. The detection strategies were categorized and summarized in terms of how to capture the EVs at the SERS substrate. In the case of the conventional label-free SERS detection strategy, it could be categorized into a dry method and a liquid state method. In the case of using the dry method, a signal of EVs was obtained by drying the EVs solution on the substrate. This method has the advantage of being able to close the physical distance of EVs. Unfortunately, in the case of using a simple substrate, the problem is that it is difficult to measure even signals due to the non-uniformity of the substrate itself and the coffee-ring effect that occurs when the EVs solution dries. To solve this problem, many studies have produced a substrate having an even electromagnetic field amplification characteristic, and among them, structures capable of stably confining EVs such as a Bee-hive structure have also been proposed. In addition, in the case of the liquid state method, which was attempted as part of the liquid biopsy, label-free SERS signals could be obtained using quantitative nanoparticle coating of individual EVs. These methods succeeded in obtaining SERS signals of weak EVs but these did not solve the heterogeneous signal of EVs. Single EV detection technology is emerging as a technology that can solve this problem, and the main technology among them is a method using Raman Spectroscopy Optical tweezers. Therefore, if this is applied to SERS, it will be possible to use it as a precision medical technology suitable for liquid biopsy. Also, the method by which signals were obtained and interpreted was discussed in aspects of classification. Notable progress in dealing with the complex and multivariate signals of EVs are then presented. In particular, it is expected that the analysis method of SERS signal through machine learning, not the existing statistical analysis method, can solve the heterogeneous by analyzing the SERS pattern of EVs, unlike peak-based analysis. It is expected that the development of various detection techniques and progress in computer science will accelerate the advancement of label-free SERS detection of EVs.

## Figures and Tables

**Figure 1 molecules-25-05209-f001:**
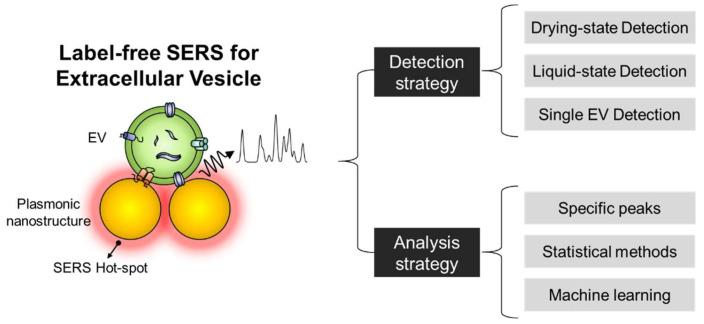
Label-free surface-enhanced Raman spectroscopy (SERS) of extracellular vesicles (EVs). The label-free SERS approach can be categorized into two major strategies of signal detection and analysis.

**Figure 2 molecules-25-05209-f002:**
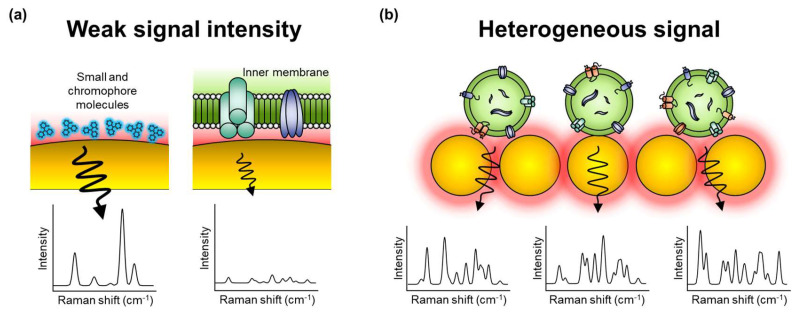
Challenges in label-free SERS of EVs. (**a**) The intrinsic SERS signal of EVs is extremely weak in intensity than in small molecules with the chromophore. (**b**) The difference in binding status and complex composition of EVs in biological samples results in fluctuations in the signal pattern.

**Figure 3 molecules-25-05209-f003:**
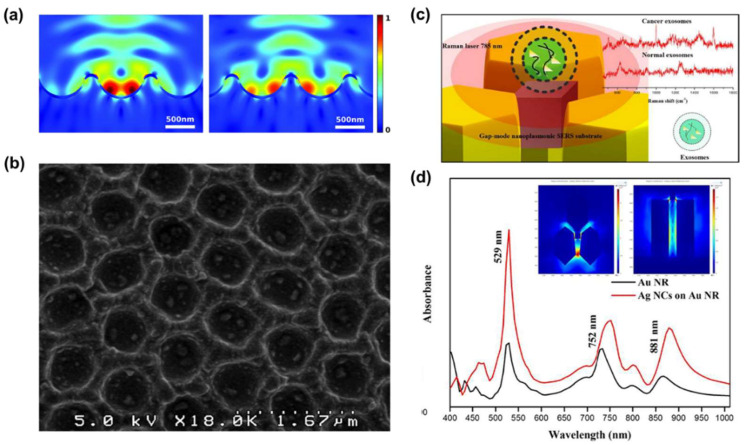
Nanostructures for the label-free SERS of dried EVs. (**a**) Electromagnetic field enhancement factor (EFEF) simulation results of the 3D plasmonic nanobowl when the laser is focused on the center of the nanobowl (left) and the outer wall of the nanobowl (right). (**b**) Scanning electron microscopy (SEM) image of a nanobowl platform with 100 nm polystyrene beads trapped inside the bowls. Reproduced from Ref. [29] with permission from The Royal Society of Chemistry. (**c**) Schematic concept of bimetallic plasmonic gap-mode SERS. (**d**) Surface plasmon resonance spectrum of the AuNR and AgNC-AuNR structures. The inset images show the local EM field distribution of bimetallic nanogaps (left: top view, right: side view). Adapted from Ref. [49], Copyright 2017, with permission from Elsevier.

**Figure 4 molecules-25-05209-f004:**
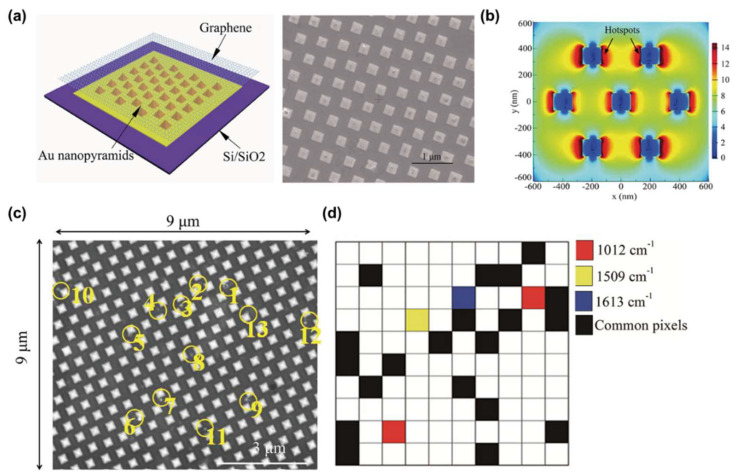
Hybrid SERS substrate-based EVs detection. (**a**) The schematic illustration and SEM image of the hybrid SERS substrate. (**b**) The EM field distribution result was calculated using a finite-difference time-domain (FDTD) simulation. (**c**) SEM image of EVs attached to the hybrid SERS substrate. The yellow circles represented attached EVs within this area. (**d**) Mapping result using SERS peak of EVs. The red, yellow, and blue represent 1012, 1509, and 1613 cm^–1^ peaks in the Raman spectrum. The black pixels indicate points where all three Raman bands were detected and were regarded as containing EVs. Adapted with permission from Ref [51]. Copyright 2019 American Chemical Society.

**Figure 5 molecules-25-05209-f005:**
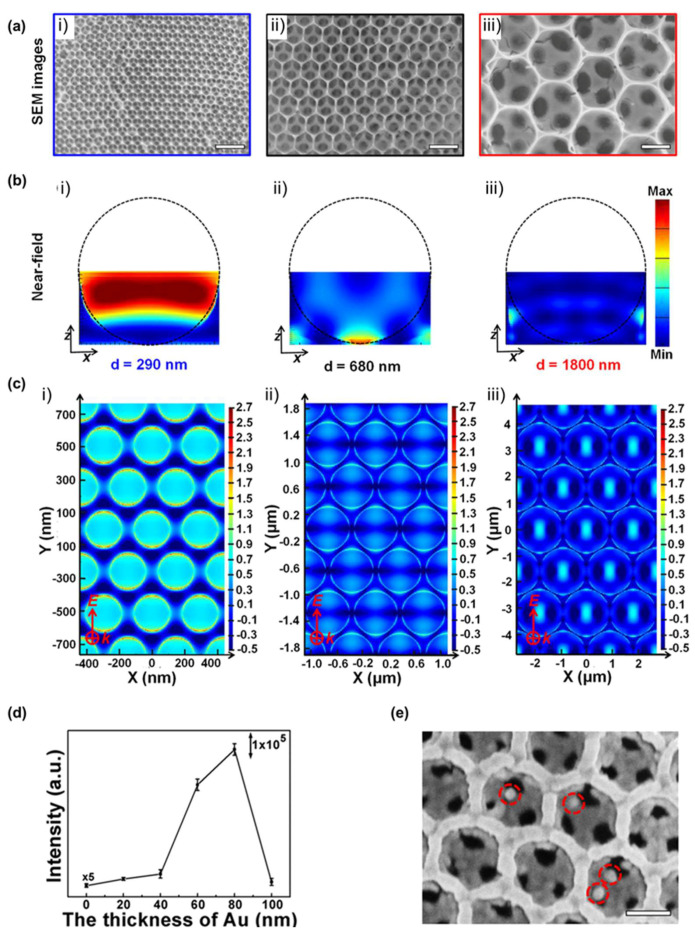
Au-coated TiO_2_ beehive structure substrate for exosome detection. (see [50]) (**a**) SEM images of the TiO_2_ MIO structure (i) 290 nm, (ii) 680 nm, and (iii) 1800 nm. The scale bars represent 1 μm. (**b**) Near-field simulation for the light trapping result at (i) 290 nm, (ii) 680 nm, and (iii)1800 nm sizes of TiO_2_ MIO structures. (**c**) FDTD simulation results for EM field enhancement at the 80 nm-thick Au-coated TiO_2_ beehive structures with pore sizes of (i) 290 nm, (ii) 680 nm, and (iii) 1800 nm. (**d**) SERS intensity of methylene blue at 1623 cm^–1^ with different thicknesses of the Au layer on the 290 nm beehive structures. (**e**) SEM image of EVs captured by the 80 nm Au-coated 290 nm TiO_2_ beehive structures. Circled areas are the EVs. The scale bar represents 300 nm. Adapted with permission from Ref [50]. Copyright 2020 American Chemical Society.

**Figure 6 molecules-25-05209-f006:**
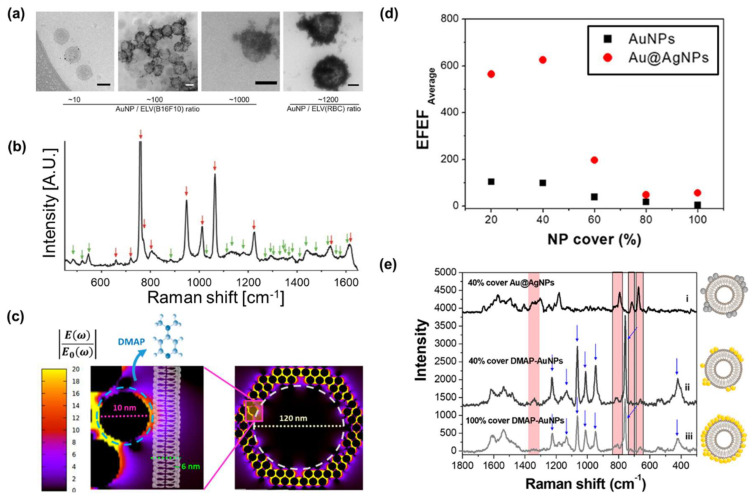
Positively charged particle-based EVs detection (**a**) TEM images of DMAP-AuNPs-coated EVs by each ratio. The scale bars represented 100 nm. (**b**) SERS measurements of DMAP-AuNP coated EVs. Red arrows indicate the DMAP-AuNP peaks, and green arrows indicate EV peaks. Ref. [56]. Copyright Wiley-VCH Verlag GmbH & Co. KGaA. Reproduced with permission (**c**) EM field enhancement results. The zoom-in image shows single DMAP-AuNP attached to the vesicle surface. The blue dashed line indicates DMAP molecules present area. (**d**) Theoretical relation between EV coverage ratio and EFEF before Ag shell-forming (black square) and after Ag shell formed (red dot). (**e**) SERS characterization of EVs coated with Ag shell-AuNPs or DMAP−AuNPs. (i) 40% coverage with Ag shell-AuNPs, (ii) 40% coverage with DMAP−AuNPs, and (iii) 100% coverage with DMAP−AuNPs. The blue arrows indicate the peaks of the DMAP. Adapted with permission from Ref. [57]. Copyright 2019 American Chemical Society.

**Figure 7 molecules-25-05209-f007:**
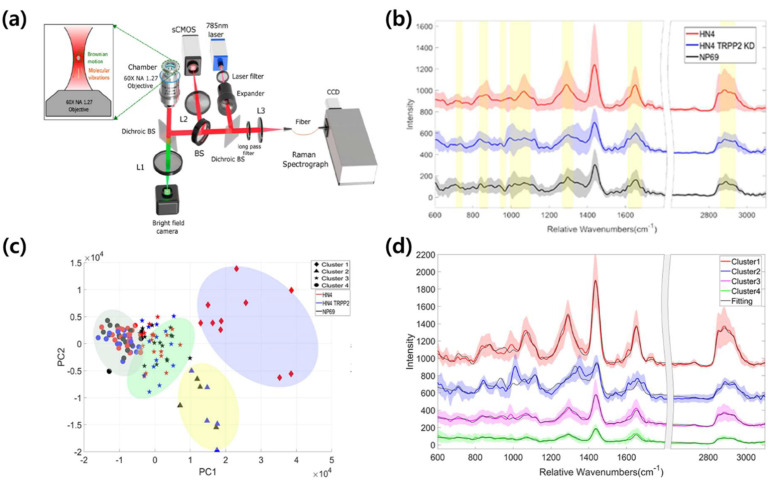
Optical tweezer-based single EV Raman detection system and the Raman signal and PCA results obtained using it. (**a**) Overview of Raman-enabled nanoparticle trapping analysis (R-NTA) system. (**b**) Raman signal of the HN4 cell line-, HN4 TRPP2 knock-down cell line-, and NP69 cell line-derived EVs. (**c**) PCA-based Raman signal clustering result. (**d**) Raman signals clustered by PCA and the average peak signal of single EVs. Adapted with permission from Ref [92]. Copyright 2020 American Chemical Society.

**Figure 8 molecules-25-05209-f008:**
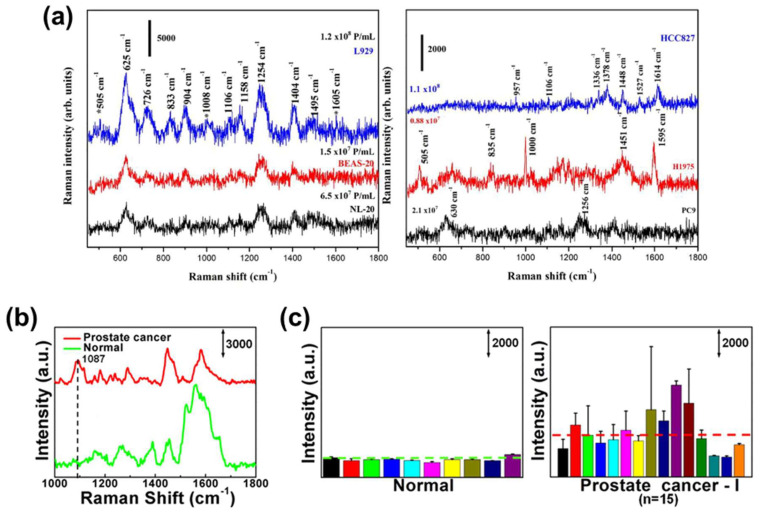
Specific bands-based EV-derived signal analysis. (**a**) The signal difference between normal lung cell line-derived EVs (NL-20, BEAS-20, and L929) and lung cancer cell lines-derived EVs (PC-9, H1975, and HCC827). Adapted from Ref. [49]., Copyright 2017, with permission from Elsevier. (**b**) The signal difference at the Raman band is associated with protein phosphorylation (P-O bond) between healthy cell- and prostate cancer cell-derived EVs. (**c**) Prostate cancer patients’ EVs tend to show higher protein phosphorylation status in the SERS result. Adapted with permission from Ref. [50]. Copyright 2020 American Chemical Society.

**Figure 9 molecules-25-05209-f009:**
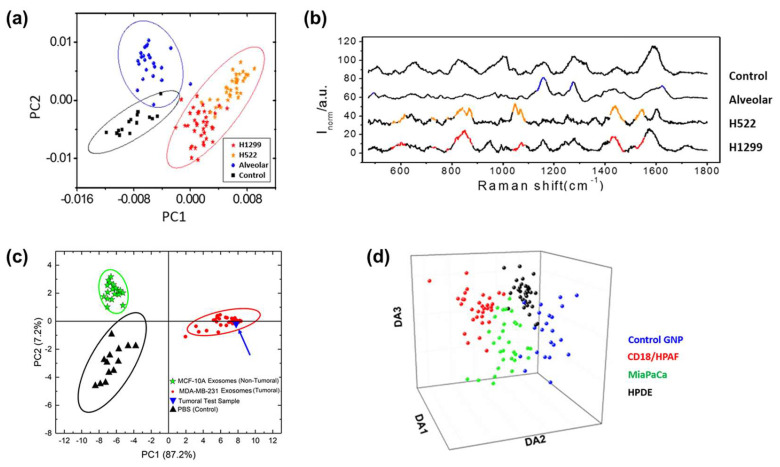
Principal component analysis (PCA) and combination with the discrimination method for EVs-SERS signal analysis. (**a**) PCA score plot of normal alveolar cell- and lung cancer cells-derived EVs. (**b**) The dominant spectral pattern of each EVs group was determined by means of PCA loading data. Adapted with permission from Ref. [59]. Copyright 2017 American Chemical Society. (**c**) Prediction of a tumoral test sample by projection to the pre-classified PCA plot. The projected spectrum of the tumoral test sample was located within the 95% confidence ellipse of tumoral EVs. Adapted with permission from Ferreira, N.; Marques, A. et al., Label-free nanosensing platform for breast cancer EVs profiling. ACS sensors 2019, 4, (8), 2073-2083. Copyright 2020 American Chemical Society. (**d**) Classification of pancreatic epithelial cell (HPDE)- and pancreatic cancer cells (MiaPaCa and CD18/HPAF)-derived EVs using PC-DFA. Adapted from Ref. [54]., Copyright 2019, with permission from Elsevier.

**Figure 10 molecules-25-05209-f010:**
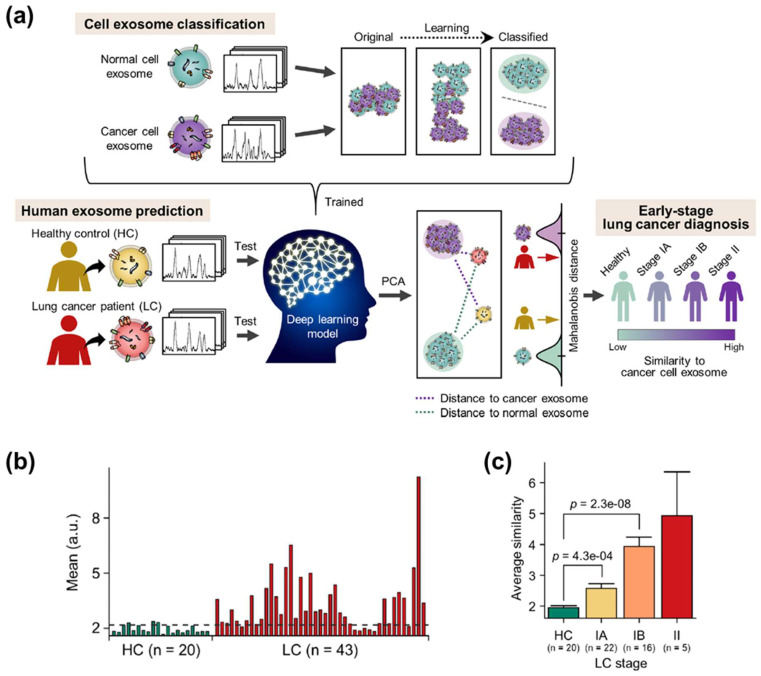
Deep learning-based early lung cancer EVs identification. (**a**) Overview of deep learning-based classification of cell lines-derived EVs and prediction of lung cancer patients using the cell EVs data. (**b**) Difference in similarity to lung cancer cell-derived EVs between healthy controls and lung cancer patients (stage I and II). (**c**) Gradual increase of the similarity value according to cancer stages.

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
