# Peer review of "Extracellular Vesicle Identification Using Label-Free Surface-Enhanced Raman Spectroscopy: Detection and Signal Analysis Strategies"

_molecules, 2020, doi:10.3390/molecules25215209_

Round 1

Reviewer 1 Report

I thank the editors for the chance to review the interesting review article by Shin, Seo, and Choi regarding the use of Label-free SERS to detect and analyze extracellular vesicles.

While other related reviews exist in the literature on plasmonic approaches applied to vesicles, or SERS applied broadly to bioanalysis, there are no reviews that focus specifically on SERS applied to extracellular vesicles. Therefore, I find this review fills a needed gap in the literature.

However, there are several concerns with the review that lead me to be hesitant to have it published in its present form.

  1. The major concern with this paper is the English language level is not high enough for a review paper. There are several minor and major errors in grammar and terminology throughout the article that make it rather difficult to read. As the readability of a review article is at a premium compared to experimental articles, I would suggest an English language editing service or having it looked over by a native speaker prior to resubmission.
  2. The article seems too narrow. For example, why do the authors focus on exosomes compared to all extracellular vesicles? As many of the articles the authors cite are not specific to exosomes vs. microvesicles, etc. I would suggest widening the article to encompass SERS applied to extracellular vesicles generally, rather than exosomes specifically.
  3. The authors also don't discriminate between SERS methods that examine multiple vesicles at once compared to single-vesicle methods. As spontaneous Raman and SERS are both well-placed to explore the vesicle-to-vesicle variability in individual vesicles (see Yichuan Dai et al., Analytical Chemistry 92(7), 5585-5594, 2020). I would suggest to add a section specifically on individual vesicle detection.
  4. Section 3.2.3 should be moved to Section 4 and expanded. The authors' viewpoint on the challenges and future directions of SERS analysis of vesicles, as experts in the field, are one of the key additions this article would make to the literature. Therefore, the current challenges in vesicle purification, capture, measurement, and analysis, and the authors' view on the promising solutions to those problems, should be the main focus of the conclusions section.
  5. The authors' citation of references in the figure captions leaves it ambiguous whether they have actually received permission from the original publishers to re-use this material. The copyright issues (if any) should be resolved before publication.

Once these deficiencies have been resolved, the manuscript is likely to be a well-received and timely review article in the literature.

Author Response

Thanks for your kind review. We prepared revision letters for our response.

please check our revision letter.

Sincerely,

Yeonho Choi, PhD

Reviewer 2 Report

Overall this review represents a modest slice of the current SERS research without any true value beyond a shallow introduction to the topic. The authors do not dive deeply into any of the cited papers, or offer any critical review or perspective beyond vague, brief summaries of just some of the recent literature. Ultimately it offers no practical benefit compared to the many recent reviews on plasmonics/Raman for EV analysis. There is a small promising section at the end of the manuscript (just 2 paragraphs) that begins to describe some challenges and prospects, but again falters to vague generalities and light overviews of a minimal number of recent studies. A review should offer some critical aspect of the field to be of use in and of itself. It should synthesize information across studies and make connections, not simply copy and paste the major conclusions without deeper analysis. 

Though there are many examples of how this review falls short of the mark, here is one practical example: on page 13, line 413, the authors state: "A method capable of selectively isolating and detecting disease cell-derived exosomes can be a practical approach for diagnosis, but a feasible approach is yet unclear.” There are literally dozens of papers published in the last few years demonstrating a wide variety of methods capable of isolating disease-associated EVs. How appropriate are these methods for downstream SERS? That is just one example of an unexplored angle set up by this manuscript but not delivered.

Another point is that the term "exosome" is used inappropriately in the title and throughout. Exosomes refer to the canonical population of EVs released via the multivesicular body cellular machinery, and are now known to be only a small fraction of the total EV load detected in biofluids. Importantly, there are no distinguishing markers or other characteristics that can identify exosome-type EVs amongst the larger isolate, so it is no longer correct to refer to broadly isolated particles strictly as exosomes. The authors should refer to the most recent suggested standards set by the International Society for Extracellular Vesicles (ISEV) to address the nomenclature. While it may be permissible to call them “exosomes” in special circumstances, there should be an explicit reason for doing so clearly stated in the manuscript. This comment represents a larger point in this manuscript, the description of the vesicles seems out of date with the modern literature, and again too shallow. It is now known that there are many EV subpopulations within the nano-sized vesicles described herein, including ones varying by size (e.g. less than 50 nm, between 50-100nm, and so on) and density (low vs high). There are also ectosomes, microvesicles, exomeres, etc., which are not considered. Therefore it is unlikely that this review would be useful to the target audience of EV researchers. 

Author Response

(The authors gave the same response as above.)

Round 2

Reviewer 1 Report

I'm grateful for the opportunity to review the review article by Shin and co-authors. The paper is improved from the previous version, but serious issues still remain.

  1. Regarding single-vesicle detection methods, the authors chose to focus on spontaneous Raman methods, which does not fit with the theme of their review article, making this section appear strange and out of place. There are several papers exploring single EV SERS measurements, or where vesicles could be examined on a one-by-one basis (the honeycomb and nanovoid structures introduced earlier by the authors are one example). The fact that the authors did not pull out these single vesicle papers from the SERS literature speaks to the other reviewer's concern that this review did not probe the literature deep enough to be comprehensive.
  2. The conclusion of the review article should be much longer, presenting the authors' overall view of the state of the field. What are the key remaining challenges, where should new researchers direct their effort, where do SERS methods fit in with other plasmonic approaches like SPR, or other optical approaches such as spontaneous Raman or fluorescence?
  3. There are still several English language issues, although these have been somewhat improved from the previous version.

As the paper currently stands, I would have to agree with the other reviewer that this review will be unlikely to make a substantial impact in the field. However, with an earnest revision where the literature is explored deeper, placed into better context, and the authors' viewpoints and opinions are more clearly and comprehensively stated, the review could be acceptable for publication.

Author Response

Please, please check our Secondary revision letter.

Reviewer 2 Report

I commend the authors for addressing the weaknesses I identified in the previous draft. I believe that the current draft has risen to the level to warrantees publication. However, prior to this, it must be completely revised for improving the English language. As it stands, this is the major obstacle to publishing. The authors mentioned that they believed they addressed this in this new version, but that is not the case. I recommend hiring a specialist firm who deals in this to revise the paper.

Author Response

Please, check our Secondary revision letter.
